# Impaired Bile Acid Metabolism and Gut Dysbiosis in Mice Lacking Lysosomal Acid Lipase

**DOI:** 10.3390/cells10102619

**Published:** 2021-10-01

**Authors:** Vinay Sachdev, Madalina Duta-Mare, Melanie Korbelius, Nemanja Vujić, Christina Leopold, Jan Freark de Boer, Silvia Rainer, Peter Fickert, Dagmar Kolb, Folkert Kuipers, Branislav Radovic, Gregor Gorkiewicz, Dagmar Kratky

**Affiliations:** 1Gottfried Schatz Research Center, Molecular Biology and Biochemistry, Medical University of Graz, 8010 Graz, Austria; v.v.sachdev@amsterdamumc.nl (V.S.); madalina.dutamare@gmail.com (M.D.-M.); m.korbelius@medunigraz.at (M.K.); nemanja.vujic@medunigraz.at (N.V.); leopold_christina@yahoo.co.uk (C.L.); silvia.rainer@medunigraz.at (S.R.); branislav.radovic@medunigraz.at (B.R.); 2Department of Pediatrics, University Medical Center Groningen, 9713 Groningen, The Netherlands; j.f.de.boer@umcg.nl (J.F.d.B.); f.kuipers@umcg.nl (F.K.); 3Department of Laboratory Medicine, University Medical Center Groningen, 9713 Groningen, The Netherlands; 4Division of Gastroenterology and Hepatology, Department of Internal Medicine, Medical University of Graz, 8010 Graz, Austria; peter.fickert@medunigraz.at; 5Cell Biology, Histology and Embryology, Gottfried Schatz Research Center, 8010 Graz, Austria; dagmar.kolb@medunigraz.at; 6Center for Medical Research Medical University of Graz, 8010 Graz, Austria; 7Diagnostic and Research Institute of Pathology, Medical University of Graz, 8010 Graz, Austria; gregor.gorkiewicz@medunigraz.at; 8BioTechMed-Graz, 8010 Graz, Austria

**Keywords:** lysosomal acid lipase, Western-type diet, cholesterol absorption, FGF15, gut microbiota

## Abstract

Lysosomal acid lipase (LAL) is the sole enzyme known to be responsible for the hydrolysis of cholesteryl esters and triglycerides at an acidic pH in lysosomes, resulting in the release of unesterified cholesterol and free fatty acids. However, the role of LAL in diet-induced adaptations is largely unexplored. In this study, we demonstrate that feeding a Western-type diet to Lal-deficient (LAL-KO) mice triggers metabolic reprogramming that modulates gut-liver cholesterol homeostasis. Induction of ileal fibroblast growth factor 15 (three-fold), absence of hepatic cholesterol 7α-hydroxylase expression, and activation of the ERK phosphorylation cascade results in altered bile acid composition, substantial changes in the gut microbiome, reduced nutrient absorption by 40%, and two-fold increased fecal lipid excretion in LAL-KO mice. These metabolic adaptations lead to impaired bile acid synthesis, lipoprotein uptake, and cholesterol absorption and ultimately to the resistance of LAL-KO mice to diet-induced obesity. Our results indicate that LAL-derived lipolytic products might be important metabolic effectors in the maintenance of whole-body lipid homeostasis.

## 1. Introduction

As cholesterol is a major component of biological membranes and a substrate for the generation of steroid hormones and bile acids, its synthesis and uptake are tightly regulated [1]. Cholesterol and triglycerides (TG) transported by apolipoprotein B-containing lipoproteins (i.e., chylomicron (CM) remnants and low-density lipoproteins (LDL)) are taken up into the cell by receptor-mediated endocytosis and processed in lysosomes [2]. Therefore, the lysosome is a critical sorting hub for lipoprotein-derived cholesterol. Lysosomal acid lipase (LAL), encoded by the *Lipa* gene, is to date the sole lipid hydrolase known to be involved in the degradation of cholesteryl esters (CE), TG, diacylglycerol, and retinyl esters in the lysosomal lumen. The critical importance of LAL-mediated lipid processing is evident in patients suffering from LAL deficiency (LAL-D). Disease severity varies largely depending on the type of mutation and is determined by the absence or presence of residual LAL activity, leading to either Wolman disease (WD) or CE storage disease (CESD), respectively. Whereas patients affected by WD are unlikely to survive beyond 6 months of age predominantly due to malabsorption and failure to thrive, CESD patients can reach adulthood but suffer from severe dyslipidemia, accelerated atherosclerosis, early cardiovascular events, and liver failure [3]. LAL-D is a rare disorder with an estimated overall disease prevalence of 1:40,000 to 1:300,000, depending on ethnicity, geographical location, and sources [4,5,6,7]. In addition to hepatosplenomegaly and dyslipidemia (in 74–90% of patients), gastrointestinal symptoms such as malnutrition, cachexia, diarrhea, steatorrhea, and vomiting were described in 30% of 206 adult and pediatric patients [5,7,8,9]. The approval of enzyme replacement therapy in 2015 dramatically changed the treatment strategy for LAL-D from supportive care to sustained improvement in the clinical outcomes, although with some therapeutic and significant pharmacoeconomic limitations [10].

Human and mouse LAL share 75% identity and 95% amino acid sequence similarity, making LAL-knockout (LAL-KO) mice a highly suitable model system to study the mechanistic and physiological roles of LAL [11]. LAL-KO mice reflect a CESD-like phenotype with dyslipidemia, shortened lifespan, and excessive accumulation of CE and TG in the liver, spleen, and small intestine [12]. LAL-derived fatty acids (FA) are a critical source of precursors for the synthesis of lipid mediators [13]. We and others have shown that LAL is critical for the maintenance of FA metabolism and overall energy homeostasis [14,15]. In the livers of LAL-KO mice, reduced FA availability leads to impaired very-low-density lipoprotein (VLDL) secretion with concomitant improved insulin sensitivity and glucose tolerance [16]. Hepatocyte-specific loss of LAL is sufficient to trigger hypercholesterolemia, hepatic inflammation, and cholesterol accumulation in the liver [17].

The liver plays a central role in maintaining cholesterol homeostasis by balancing multiple pathways, including dietary cholesterol uptake, de novo cholesterol, and bile acid synthesis, lipoprotein synthesis, biliary cholesterol excretion, and reverse cholesterol transport. Cholesterol is largely excreted from the body after biochemical modification to bile acids (BA) and steroid hormones [18,19]. Cholesterol 7α-hydroxylase (CYP7A1) catalyzes the first and rate-limiting step in the classical BA synthesis pathway. Newly synthesized BA is stored in the gallbladder and released postprandially into the intestinal lumen to emulsify dietary lipids. The majority of BA (~95%) is reabsorbed in the terminal ileum via the apical sodium-dependent bile salt transporter (ASBT) [18,20]. Enterohepatic BA homeostasis is controlled by the farnesoid X receptor (FXR) via induction of mouse intestinal fibroblast growth factor 15 (FGF15; human ortholog FGF19), which suppresses hepatic CYP7A1 expression as an endocrine signal with negative feedback [21,22].

BA signaling is a tightly regulated process, which can be influenced by a variety of factors. The physicochemical characteristics of individual BA influence the capacity for lipid emulsification and the general signaling properties of the biliary pool [18,23]. The physiological effects of altered BA composition in regulating cholesterol excretion in mouse models have recently been described [24]. Gut microbiota and BA composition are interdependent; intraluminal microbial BA modulation through deconjugation and dehydroxylation processes determines the composition of secondary BA, while BA-specific bacteriostatic effects regulate the gut microbial ecosystem [25,26]. Furthermore, certain factors such as dietary lipid content may simultaneously regulate both the size of the BA pool and the composition of the gut microbiome [26,27,28].

This study shows that LAL-KO mice fed a high-calorie diet (Western-type diet, WTD) display profound changes in enterohepatic BA metabolism and the intestinal microbiome compared to wild-type (WT) mice. An altered BA composition potentially hinders nutrient absorption and increases fecal lipid excretion. The overall metabolic adaptations result in attenuated diet-induced weight gain but exacerbated dyslipidemia in LAL-KO mice, highlighting the importance of LAL-derived lipolytic products in maintaining gut-liver crosstalk.

## 2. Materials and Methods

### 2.1. Animals and Diets

Age-matched male LAL-KO mice and their corresponding WT littermates [12] on a C57BL/6J background [16] were used for all experiments unless otherwise indicated. Mice had *ad libitum* access to water and food and were maintained under a 12 h light/12 h dark cycle in a temperature-controlled environment. Mice were fed a standard chow diet (Altromin 1324, Lage, Germany), after which the animals were challenged with a Western-type diet (WTD) (TD88137; 21% fat, 0.2% cholesterol; Ssniff Spezialdiaeten GmbH, Soest, Germany) for 2–6 weeks. All experiments were performed in accordance with the European Directive 2010/63/EU and approved by the Austrian Federal Ministry of Education, Science and Research (Vienna, Austria; BMWFW-66.010/0065-WF/V/3b/2015, BMWFW-66.010/0081-WF/V/3b/2017, BMBWF-66.010/0106-V/3b/2019; 2020-0.129.904).

### 2.2. Plasma Lipid Parameters, Lipoprotein Profiles, and 7α-hydroxy-4-cholesten-3-one (C4)

Plasma lipid parameters, lipoprotein profiles after separation by fast-protein liquid chromatography, and C4 concentrations were determined as previously described [29,30].

### 2.3. Analysis of Circulating FGF15 Concentrations

WT and LAL-KO mice fed a WTD for 2 weeks were fasted for 6 h and gavaged with 200 µL corn oil. Ninety minutes post-gavage, blood was collected, and plasma was isolated by centrifugation at 5200× *g* for 7 min at 4 °C. Plasma FGF15 concentrations were measured by ELISA according to the manufacturer’s protocol (R&D Systems, Minneapolis, MN) [31].

### 2.4. RNA Isolation, Reverse Transcription, and Quantitative Real-Time PCR

RNA was isolated from tissues harvested from 6 h-fasted WT and LAL-KO mice fed a WTD for 6 weeks. Two micrograms of RNA were reverse transcribed using the High-Capacity cDNA Reverse Transcription Kit (Applied Biosystems, Carlsbad, CA, USA). Three microliters of diluted cDNA (1:50) and 1 μL of each forward and reverse primer (Appendix A) were mixed with 5 μL QuantiFast SYBR Green master mix (Qiagen, Hilden, Germany). Samples were analyzed in duplicate and normalized to the expression of peptidylprolyl isomerase A (*Ppia*, also known as *cyclophilin A*) as a housekeeping gene. Real-time PCR was performed on a Roche LightCycler 480 (Roche Diagnostics, Palo Alto, CA, USA). Expression profiles were calculated using the 2^−ΔΔCt^ method.

### 2.5. Western Blotting Analysis

Samples were lysed in RIPA buffer, and protein concentrations were quantitated (DC™ Protein assay, Bio-Rad Laboratories, Hercules, CA, USA). Lysates (40 µg protein) were separated by SDS-PAGE and transferred onto PVDF membranes. Non-specific binding sites of the membranes were blocked for 1 h at room temperature (5% solution of milk powder or 1% BSA in washing buffer). For detection of the proteins of interest, we used polyclonal antibodies against pERK (#9106) and ERK (#4695) (both 1:1000; Cell Signaling Technology, Danvers, MA), CYP7A1 (ab65596), and TFEB (ab2636) (both 1:1000; Abcam, Cambridge, United Kingdom). Polyclonal anti-rabbit calnexin (1:1000; Santa Cruz, Heidelberg, Germany), β actin (1:10,000; Merck KGaA, Darmstadt, Germany), or HDAC1 (#2062, 1:1000; Cell Signaling Technology) were used as loading controls. HRP-conjugated goat anti-rabbit (1:2500) and rabbit anti-mouse antibodies (1:500) (Dako, Glostrup, Denmark) were visualized by enhanced chemiluminescence detection on a ChemiDocTM MP imaging system (Bio-Rad Laboratories).

### 2.6. Electron Microscopy

The small intestine in LAL-KO mice accumulates excess lipids, specifically in the proximal part [6]. Freshly harvested duodena from chow diet-fed mice in the fed state were immediately fixed in 2.5% (wt/vol) glutaraldehyde and 2% (wt/vol) paraformaldehyde, buffered in 100 mM cacodylate buffer pH 7.4, and incubated at room temperature for 3 h. Post-fixation, samples were treated with 2% osmium tetroxide (diluted in 200 mM cacodylate buffer) for 2–3 h at room temperature. After washing for 2 h in 100 mM cacodylate buffer, the specimens were dehydrated in a graded series of ethanol (50%, 70%, 80%, 96%, 100% p.a.), infiltrated with propylene oxide/TAAB (Agar Scientific, Essex, Great Britain) embedding resin (propylene oxide for 1 h at room temperature, propylene oxide/TAAB 1:1 for 3 h at room temperature, propylene oxide/TAAB 1:3 o/n at 4 °C), finally embedded in pure TAAB resin, and polymerized (2 × 1.5 h, 48 °C). Sections stained with lead citrate and platinum blue were imaged at 120 kV using a Tecnai G 2 FEI microscope (FEI, Eindhoven, The Netherlands) equipped with a Gatan ultrascan 1000 CCD camera.

### 2.7. Energy Metabolism In Vivo

Energy intake and energy expenditure were assessed using a climate-controlled indirect calorimetry system (TSE Systems, Bad Homburg, Germany) as described [14]. WTD-fed WT and LAL-KO mice were housed in automatic metabolic cages at room temperature in a regular light-dark cycle (12 h light, 12 h dark) with free access to food and water. Energy expenditure was measured every 15 min.

### 2.8. Acute Cholesterol Absorption

Acute cholesterol absorption was measured as described previously [30]. Chow diet-fed mice were fasted for 4 h and thereafter gavaged with 200 µL corn oil containing 2 µCi [^3^H]cholesterol (ARC Inc., St Louis, MO, USA) and 200 µg cholesterol. Four hours post-gavage, plasma, liver, and three parts of the small intestine (duodenum, jejunum, ileum) were isolated. Intestinal tissues were rinsed with PBS to remove luminal contents before all tissues were lyophilized overnight. Radioactivity in plasma and tissues was analyzed by liquid scintillation counting.

### 2.9. Basolateral FA Uptake

FA uptake from the basolateral side of enterocytes was determined as previously described [32]. Briefly, chow diet-fed mice were fasted for 4 h and injected intraperitoneally with 100 μL intralipid (Fresenius Kabi Austria GmbH, Graz, Austria) containing 7 μCi [9,10-3H(N)]-oleate (Hartmann Analytics, Braunschweig, Germany). Radioactivity in plasma and lyophilized tissues (liver, duodenum, jejunum, ileum) was measured by liquid scintillation counting.

### 2.10. Fecal Neutral Sterol Measurements

Neutral sterols in feces of WT and LAL-KO mice fed a WTD for 4 weeks were quantified by GC as described [33,34] using 5α-cholestane as internal standard.

### 2.11. BA Measurements

BA measurements were performed in WT and LAL-KO mice fed a WTD for 4 weeks. Biliary BA concentrations were determined by (U)HPLC-MS/MS coupled to a SCIEX QTRAP 4500 MD triple quadrupole mass spectrometer and quantified using D4-labeled BA as internal standards [35]. For fecal BA measurements, BA in dried and grounded feces was methylated and trimethylsilylated prior to quantification by gas-liquid chromatography using 5ß-cholanic acid-7α,12α-diol as internal standard [36]. The hydrophobicity index (HI) was calculated as the sum of the molar fractions of individual BA multiplied by their individual HI values according to the procedure of Heuman [37]. Hydrophobicity index used: TCA, 0; Tά-MCA, −0.84, Tβ-MCA, −0.78; taurohyodeoxycholic acid, −0.37; Tω-MCA, −0.33; TUDCA, −0.27; TCDCA, 0.46; TDCA, 0.59; TLCA, 1. BA was grouped into primary and secondary BA based on previous reports [33,38]. Primary BA includes free and conjugated forms of CA, CDCA, ά-MCA, and β-MCA, whereas secondary BA includes DCA, LCA, ω-MCA, UDCA, and their conjugates.

### 2.12. Microbiota Analysis

Cecal contents of LAL-KO and control mice fed WTD for 4 weeks were subjected to quantitative 16S rRNA transcript amplifications and microbiota analysis as described earlier [39].

### 2.13. Isolation of Primary Enterocytes

Primary enterocytes from the jejunum of chow diet-fed LAL-KO and control mice were isolated as recently described [40].

### 2.14. Immunohistochemical Hematoxylin and Eosin as Well as Oil-Red O (ORO) Staining

Immunohistochemical staining was performed as previously described [30]. Tissues from 12 h-fasted mice were fixed in 4% neutral-buffered formaldehyde for 24 h and stored in 30% sucrose before cryosectioning. Sections (5 μm) were subsequently stained with Mayer’s hematoxylin and eosin as well as ORO.

### 2.15. Statistics

Statistical analyses were performed using GraphPad Prism 5.1 software. Statistically significant differences were determined by Student’s unpaired t-test with Welch’s correction (in case of unequal variances) for two group comparisons. Multiple group comparisons were calculated by two-way ANOVA followed by Bonferroni correction. Data represent mean values ± SD. Statistical significance levels were set at * *p* < 0.05, ** *p* ≤ 0.01, *** *p* ≤ 0.001.

## 3. Results

### 3.1. LAL-KO Mice Are Resistant to Diet-Induced Obesity

Compared to their WT controls, chow diet-fed LAL-KO mice exhibited reduced body weight and progressive loss of white adipose tissue (WAT) [12,16]. We speculated that feeding LAL-KO mice a high-calorie diet might induce body weight gain and compensate for the loss of adipose tissue. We chose a maximum 6-week regimen as feeding a high-calorie diet for a prolonged period has been shown to be lethal in a mouse model with a defect in lysosomal lipid processing [41]. LAL-KO mice already had lower body weight before we challenged them with WTD and the difference in weight gain increased during the 6-week feeding period (Figure 1a). The reduced weight gain in LAL-KO mice was independent of food intake, which was paradoxically 1.4-fold higher compared to WT littermates (Figure 1b). Energy expenditure was also considerably lower in LAL-KO mice (Figure 1c,d). WTD feeding failed to prevent the loss of gonadal fat, whereas the weight of the liver and proximal intestinal parts was increased (Figure 1e), as previously observed in chow diet-fed LAL-KO mice [12]. These data clearly demonstrate that LAL-KO mice are resistant to diet-induced weight gain.

### 3.2. LAL-KO Mice Exhibit Impaired Cholesterol Absorption

Consistent with the phenotype of LAL-KO mice and LAL-D patients [8,16], we found slightly increased plasma cholesterol concentrations (Figure 2a), which were due to an increase in the LDL fraction, whereas HDL-cholesterol was decreased (Figure 2b). Circulating TG concentrations were comparable to the control group (Figure 2a) due to depletion of TG in the VLDL fraction despite elevated LDL-TG (Figure 2c). Although fecal output was comparable (Figure 2d), fecal excretion of lipids (Figure 2e,f) and neutral sterols (Figure 2g) was markedly increased in LAL-KO mice.

To investigate whether cholesterol absorption might be affected in LAL-KO mice, we orally administered [^3^H]cholesterol. Plasma radioactivity tended to be lower (Figure 2h), and we observed reduced radioactivity in the duodenum, jejunum, and liver 4 h after the oral gavage (Figure 2i), indicating impaired dietary cholesterol absorption in LAL-KO mice. Analysis of possibly altered lipid receptors and transporters in isolated enterocytes revealed unchanged mRNA expression of *Abcg5/g8* but reduced *Npc1l1* mRNA (Figure 2j). We observed markedly increased mRNA expression of the plasma membrane cholesterol sensor *Scarb1*, suggesting that LAL-KO enterocytes attempt to counteract the decreased availability of free cholesterol partly by upregulation of SR-BI. These results indicate that lack of global LAL activity leads to inefficient intestinal lipid processing in LAL-KO mice.

### 3.3. LAL-KO Intestines Accumulate Lipids from the Systemic Circulation

WTD-fed LAL-KO mice accumulate lipids predominantly in the duodenum and jejunum, and the small intestine is markedly shorter compared to control mice (Figure 3a). We observed a severe intestinal accumulation of neutral lipids in LAL-KO mice (Figure 3b,c). Electron microscopy confirmed the abundance of lipid-filled lysosomes predominantly in the lamina propria (Figure 3d), which is consistent with previous reports describing in vivo models of LAL-D [12,42,43]. We have recently demonstrated the critical role of cytosolic lipases within enterocytes in the metabolism of lipids derived from the basolateral (blood) side of the small intestine [32,40]. To determine whether LAL-KO enterocytes accumulate lipid species from the basolateral membrane of enterocytes, we incorporated [^3^H]oleate into an intralipid emulsion, injected it intraperitoneally, and measured the tracer in different intestinal segments [32]. The incorporation of [^3^H]oleate instead of cholesterol offers the additional advantage that the radiotracer can be followed in both TG and CE fractions. While plasma radioactivity was comparable between genotypes (Figure 3e), LAL-KO mice showed elevated radioactivity in the duodenum (2.2-fold) and jejunum (2.6-fold) (Figure 3f). Detailed estimation of the relative distribution of lipid species in LAL-KO duodena revealed higher incorporation of basolaterally derived FA into TG and CE fractions (Figure 3g). Taken together, these results suggest that LAL-D in mice accelerates the accumulation of FA from the peripheral circulation through the basolateral side of enterocytes.

### 3.4. LAL-KO Mice Have Increased FGF15 Signaling

We next investigated whether alterations in BA metabolism may contribute to the reduced cholesterol absorption and increased fecal lipid excretion in LAL-KO mice. Therefore, we analyzed the expression and systemic concentration of fibroblast growth factor 15 (FGF15), which plays a central role in BA-driven enterohepatic signaling. In the ileum of WTD-fed LAL-KO mice, we found a 3.0-fold increase in *Fgf15* mRNA expression, whereas *Asbt*, small heterodimer partner (*Shp*), *Osta*, and *Fabp6* expression levels were unchanged (Figure 4a). Together with increased transcript levels of hepatic *Fgfr4* (6.8-fold) (Figure 4b), these findings suggested enhanced FGF15 signaling in LAL-KO mice. Almost absent *Cyp7a1*, sterol 12-alpha-hydroxylase (*Cyp8b1*), oxysterol 7-alpha-hydroxylase (*Cyp7b1*), and cytochrome P450, family 2, subfamily c, polypeptide 70 (*Cyp2c70*) mRNA expression levels indicated impaired BA synthesis through both the classical and alternative BA synthesis pathways. *Shp* gene expression was reduced, whereas *β-Klotho* was unchanged in livers of LAL-KO mice (Figure 4b). Increased ileal *Fgf15* mRNA expression resulted in elevated plasma FGF15 concentrations (1.3-fold) (Figure 4c). Consistent with almost undetectable CYP7A1 protein expression in the liver of LAL-KO mice (Figure 4d), the nuclear abundance of transcription factor EB (TFEB), an inducer of CYP7A1 in vivo [44], was markedly reduced (Figure 4e). Further analysis of the signaling cascade revealed increased protein expression of phosphorylated ERK (pERK) with comparable total ERK abundance in the liver of LAL-KO mice (Figure 4f). This finding is in line with an enhanced FGF15-mediated ERK signaling feedback loop that was shown to prevent nuclear translocation of TFEB. These alterations ultimately led to suppressed hepatic BA synthesis in LAL-KO mice, as evidenced by a 56% reduction in the BA precursor 7a-hydroxy-4-cholesten-3-one (C4) in the plasma (Figure 4g).

### 3.5. LAL-KO Mice Have Impaired BA Homeostasis

We then analyzed whether BA composition may be affected in LAL-KO mice. Consistent with mRNA expression and reduced circulating C4 concentrations, we found a lower BA content in the feces of LAL-KO mice (Figure 5a). The composition of biliary BA in LAL-KO mice was changed to contain increased β-muricholate (β-M) (Figure 5b and Appendix A) and consequently exhibited a more hydrophilic BA profile, as determined by the hydrophobicity index (Figure 5c). The composition of bile salt species in feces was shifted toward the more hydrophilic muricholates, especially β-M and deoxycholate (DC), rather than the more hydrophobic cholates (Figure 5d). This resulted in a significant reduction in the hydrophobicity index of the fecal bile salts (Figure 5e).

### 3.6. Drastic Alterations in the Gut Microbiome in LAL-KO Mice

Finally, we determined whether the metabolic alterations in LAL-KO mice are associated with changes in the microbiome. 16S rRNA sequencing followed by UniFrac-based PCoA revealed distinct clustering of the microbial communities isolated from the ceca of LAL-KO and control mice (Figure 6a). The differences in the microbiota phyla composition were caused by certain bacterial taxa with an increased relative abundance of Bacteroidetes (1.2-fold), Proteobacteria (1.2-fold), and Deferribacteres (1.4-fold), whereas Firmicutes and the cholesterol-degrading phylum Actinobacteria [45] were reduced by 26% and 70%, respectively (Figure 6b). The ratio of Firmicutes to Bacteroidetes, which significantly affects the maintenance of normal intestinal homeostasis [46,47], was 41% lower in the cecal microbiome of LAL-KO mice (Figure 6c). A more detailed analysis of the genus composition and clustering of microbial sequences according to their similarities revealed a highly variable abundance of operational taxonomic units (OTU). The relative abundance of Lachnospiraceae (−44%), Lactobacillales (−47%), Bacteroidales_unclassified (−36%), Erysipelotrichaceae (−99%), Alcaligenaceae (−37%), Coriobacteriaceae (−51%), and Bifidobacteriaceae (−87%) was decreased, whereas Bacteroides (1.6-fold), Porphyromonadaceae (1.6-fold), Rumminococcaceae (1.3-fold), Helicobacteraceae (2.4-fold), Prevotellaceae (2.2-fold), and Odoribacteraceae (3.3-fold) was increased in the ceca of LAL-KO mice (Figure 6d). By applying PICRUSt to our data, we were able to determine the contribution of each OTU to the total gene content of each sample. Metagenomic modeling by PICRUSt revealed several significantly downregulated KEGG pathways (Figure 6e). Remarkably, we observed a pronounced shift in signaling pathways of genes involved in BA metabolism that were almost undetectable in LAL-KO ceca (Figure 6f). Thus, these results suggest that the altered gut microbiome might be responsible for the impaired BA metabolism in WTD-fed LAL-KO mice.

## 4. Discussion

In this study, we have demonstrated that LAL critically impacts biliary homeostasis in mice fed a WTD. The main findings from our study point to several adaptations in LAL-KO mice that culminate in excessive excretion of lipids.

Intracellular lipoprotein trafficking and catabolism are dependent on LAL-mediated hydrolysis of lipoproteins internalized via receptor-mediated endocytosis [48]. A hallmark in LAL-D patients is dyslipidemia, which was suggested to be ameliorated by statin treatment [8,49]. In agreement with data from both LAL-D patients and chow diet-fed LAL-KO mice [8,16,49], WTD-fed LAL-KO mice exhibited reduced HDL-cholesterol but highly increased LDL-cholesterol concentrations. VLDL-TG levels, however, were drastically reduced, consistent with our previous report on the critical role of LAL in the regulation of VLDL secretion [16].

VLDL synthesis and secretion, LDL uptake as well as de novo lipogenesis represent major functions of the liver, with minor involvement of the intestine [50]. Intestinal lipid accumulation as a result of lipid-laden macrophages in the lamina propria is a characteristic feature of LAL-KO mice [12,16]. Consistently, we observed lipid-rich vacuoles in the intestinal lamina propria of WTD-fed LAL-KO animals. Intraperitoneal administration of [^3^H]oleate, mimicking FA uptake from the basolateral side of enterocytes, revealed an increased incorporation of radioactivity into TG in the duodenum of LAL-KO mice. We have recently shown that adipose triglyceride lipase (ATGL) and its coactivator CGI-58 are critical for processing a specific pool of reabsorbed TG within the enterocyte. These lipids originate from the basolateral absorption in enterocytes and are not destined for chylomicron synthesis [32,40]. Taken together, these data support the essential role of lipases in the processing of reabsorbed TG in the intestine. While the exact role and molecular mechanisms of (intestinal) LAL in this process are still ambiguous, future studies should determine whether LAL participates in the processing of lipids delivered apically to enterocytes.

It has been recently described that reverse cholesterol transport in LAL-KO mice is reduced [51]. Furthermore, we have demonstrated that hepatocyte-specific loss of LAL does not regulate fecal lipid balance [17]. The increased fecal neutral sterol loss and fecal lipid excretion, together with reduced CYP7A1 concentrations in our study, in part, explained the modulation of intestinal cholesterol absorption in LAL-KO mice. It has been previously shown that feeding a high-cholesterol diet to NPC1-KO mice with impaired lysosomal cholesterol release resulted in reduced BA secretion [52]. In line, the FGF15-CYP7A1 axis was altered in WTD-fed LAL-KO mice. Increased ileal *Fgf15* transcript and FGF15 plasma levels resulted in an upregulation of the downstream ERK signaling pathway and repression of hepatic CYP7A1. A defective lysosomal feedback loop affecting gut-liver BA metabolism has recently been described [44]. Similar to our study, the authors found that FGF15/19 reduces nuclear TFEB transport, resulting in downregulation of hepatic CYP7A1 in an SHP-independent manner [44]. As a regulator of *Lipa* expression, TFEB modulates LAL activity [53]. However, whether positive feedback regulation by LAL may reciprocally drive TFEB concentrations remains elusive. Since TFEB is a cellular nutrient- and stress-sensing transcription factor [53,54], it is plausible that CE accumulation in LAL-D lysosomes may impair TFEB nuclear translocation. Although a previous report highlighted this possibility in the context of lysosomal sphingosine accumulation [55], further studies are needed to decipher the effect of TFEB on cholesterol homeostasis.

We have confirmed that reductions of *Cyp7a1*, *Cyp8b1*, *Cyp7b1*, and *Abcb11* were causally related to altered BA synthesis in LAL-KO mice. Recent reports underscored that both BA pool size and composition are of paramount importance for cholesterol homeostasis [24,56,57]. The FXR/FGF15/19 gut-liver axis has been reported to cause a hydrophilic BA pool, with a high muricholate/cholate ratio, which enhances sterol excretion and thus appears to be of significant physiological relevance in vivo [24]. CYP8B1 mediates BA 12α-hydroxylation. Hence, the CYP7A1/CYP8B1 ratio determines the balance between cholate and chenodeoxycholate/muricholate species between the genotypes [21]. The higher muricholate levels in LAL-KO mice are most likely also due to less CA-derived BA and relatively more CDCA-derived BA. Comparable to Cyp2c70^+/-^ mice, our data suggest that the reduced amount of Cyp2c70 present in LAL-KO mice is still sufficient to efficiently convert (nearly all) CDCA produced into MCA [34]. A lower biliary and fecal hydrophobicity index and a corresponding increase in fecal sterol excretion supported the finding of altered BA synthesis in LAL-KO mice. It should be noted that the extrapolation of our results in LAL-KO mice to LAL-D patients is limited due to the inherent differences in BA synthesis between mice and humans. Hydrophobic chenodeoxycholate is efficiently hydroxylated to hydrophilic muricholates in the livers of mice, a reaction that does not occur in humans. Thus, a possible increased FGF19 content in LAL-D patients could lead to a more hydrophobic BA pool, theoretically resulting in decreased fecal sterol excretion and exacerbated dyslipidemia.

Regulation of BA metabolism and the gut microbiota are highly interdependent. In addition to facilitating intestinal lipid uptake, which contributes to energy mobilization, BA also modulate intestinal microbial proliferation and diversity [25,26,56]. Conversely, disturbances in the composition of the gut microbiota strongly regulate the size and composition of the BA pool [38,47,48]. The intestinal microbiota alters BA architecture mainly by controlling its transformation via deconjugation and dehydroxylation steps. We observed a pronounced decrease in the relative cecal population of Firmicutes and Actinobacteria, while the abundance of Bacteroidetes was significantly increased in WTD-fed LAL-KO mice. An inverse correlation of such microbiota patterns in murine models and humans is causally associated with diet-induced obesity since obese humans and mice showed a higher ratio of Firmicutes to Bacteroidetes compared to their lean counterparts [26,58,59,60]. Thus, the alterations in the major phyla in the gut microbiota may partially confer resistance to diet-induced weight gain in LAL-KO mice. In addition, the increased biliary deoxycholic acid excretion observed in LAL-KO mice could also be in part attributed to gut microbiome changes, as increased Bacteroidetes and reduced Firmicutes abundance were described in mouse models with higher deoxycholic acid concentrations [59,61]. Moreover, the considerably reduced Lactobacillus genus may additionally influence the phenotype of WTD-fed LAL-KO mice. Lactobacilli are involved in the regulation of bile salt hydrolase activity in the mouse intestine [62], responsible for deconjugation of conjugated BA such as tauro-β-muricholic acid and host energy metabolism [47,63]. It is plausible that increased muricholic acid concentrations in LAL-KO mice are (at least in part) a consequence of gut dysbiosis. In this context, it is noteworthy that increased β-muricholic acid, as well as reduced Firmicutes and Lactobacilli levels, were associated with intestinal FXR antagonism, such as reduced ileal FGF15 expression in mice [47,60]. Conversely, intestinal FXR overexpression or FGF19 administration in intestinal-specific FXR-KO mice was sufficient to induce a shift in BA composition from cholate to muricholate, resulting in higher BA hydrophilicity a reduction in CYP7A1 expression, and an increase in fecal neutral sterols [24,64]. Of note, these studies were performed with either FXR-targeted pharmacological approaches or genetically modified mouse models that induce supraphysiological alterations in intestinal FXR expression. Whether modulation in intestinal FXR expression induced after feeding a high-calorie diet would follow similar paradigms remains unknown [65]. Our findings that FGF15 and hydrophilic muricholates are simultaneously increased in WTD-fed LAL-KO mice can be reconciled with the above studies by postulating that BA changes are in part associated with altered microbiome composition. Of note, LAL-KO mice phenocopy the major clinical manifestations of CESD but not WD (e.g., diarrhea, cachexia, or failure to thrive). Therefore, although our data provide valuable insight into high-calorie feeding in our mouse model, it is possible that disease severity is higher in LAL-D patients. It may be interesting to investigate whether the current findings can be applied to other models of lysosomal storage diseases that also exhibit dyslipidemia, inflammatory responses, and neurodegenerative pathogenesis. The limitation of the present study is highlighted by the associative nature of the results linking LAL-D to gut dysbiosis and alteration of BA homeostasis. Future studies are warranted to examine the precise host responses to LAL using fecal transplantation experiments in global and tissue-specific LAL-D mouse models.

While the molecular basis of LAL-FGF15 regulation is currently unclear, we postulate that metabolic adaptations in the LAL-D intestine limit lipid absorption and hence promote fecal lipid loss under WTD feeding. We speculate that these intestinal adaptations likely serve to protect LAL-KO cells, already stressed by lipid accumulation, from additional lipotoxic effects of dietary lipids.

## Figures and Tables

**Figure 1 cells-10-02619-f001:**
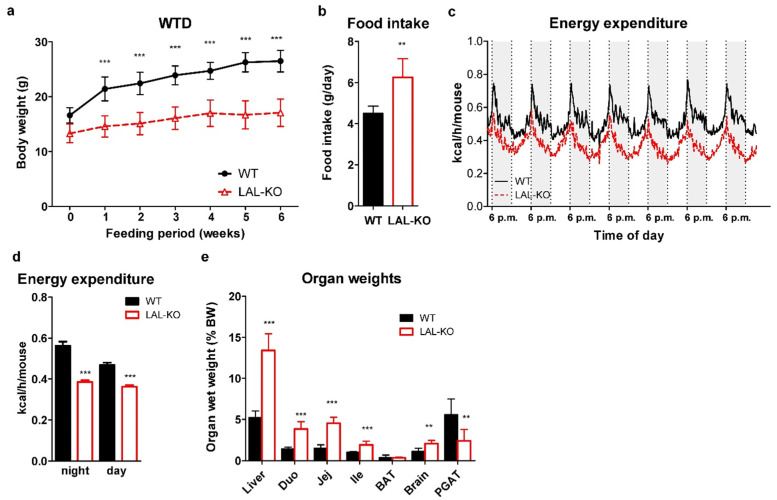
Resistance to diet-induced obesity and altered energy metabolism in LAL-KO mice: (**a**) Body weight of 12-week-old male mice during a WTD feeding period of 6 weeks and (**b**) daily food intake. (**c**,**d**) Energy expenditure measured by indirect gas calorimetry in WTD diet-fed WT (n = 6, black line) and LAL-KO mice (n = 6, red line); shaded areas represent dark phase (6 p.m.–6 a.m.); non-shaded, light phase (6 a.m.–6 p.m.). (**e**) Organ weights relative to body weight (Duo, duodenum; Jej, jejunum; Ile, ileum; BAT, brown adipose tissue; PGAT, perigonadal adipose tissue; n = 6). Data represent means ± SD; n = 6; *p* ≤ 0.01 (**), *p* ≤ 0.001 (***). (**a**) ANOVA; (**b**,**d**) Student’s unpaired t-test.

**Figure 2 cells-10-02619-f002:**
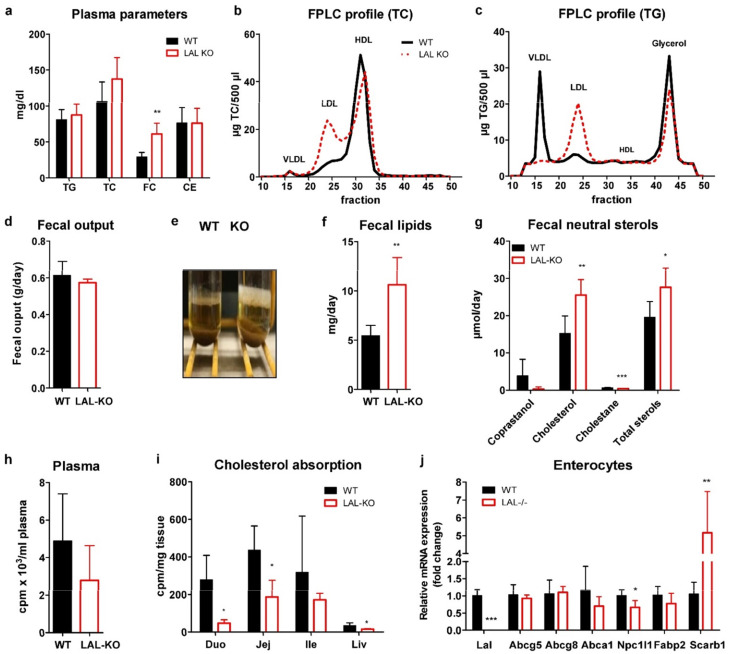
Impaired cholesterol absorption in LAL-KO mice: (**a**) Plasma lipid parameters and lipoprotein profiles of (**b**) TC and (**c**) TG concentrations after separation by fast performance liquid chromatography of pooled plasma from 12 h-fasted male mice (n = 6, 25 weeks old, 6 weeks on WTD). (**d**) Daily fecal output. (**e**) Feces of WTD-fed male mice (n = 6, 12–14 weeks old) were collected for 72 h. The picture shows the upper lipid layers in the extraction tubes of fecal samples. Quantification of (**f**) total fecal lipids and (**g**) fecal neutral sterols. (**h**,**i**) Cholesterol absorption was measured in chow diet-fed female mice (n = 4–5, 10 weeks old). After a 4 h-fasting period, mice were gavaged with 200 μL corn oil containing 2 μCi [^3^H]cholesterol and 200 μg cholesterol. Radioactivity was measured 4 h post-gavage in (**h**) plasma and (**i**) isolated tissues by liquid scintillation counting. (**j**) Enterocyte mRNA expression of cholesterol transporters relative to *Gapdh* as reference gene (n = 5–6). Data represent means + SD; *p* < 0.05 (*), *p* ≤ 0.01 (**), *p* ≤ 0.001 (***). Student’s unpaired t-test.

**Figure 3 cells-10-02619-f003:**
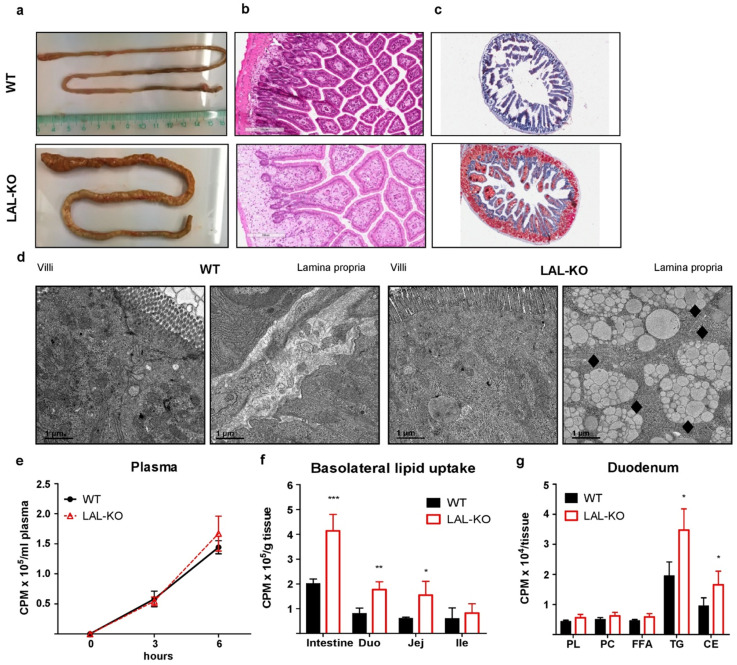
Intestines of LAL-KO mice accumulate lipids derived from the systemic circulation: (**a**) Representative photographs of the small intestine, micrographs of (**b**) H&E, and (**c**) oil red O-stained duodena from male WT and LAL-KO mice (17 weeks of age). (**d**) Representative duodenal electron micrographs; scale bar, 1 µm; black symbols indicate lipid-filled lysosomes in the lamina propria. Radioactivity in (**e**) plasma, (**f**) small intestine, and (**g**) absolute lipid distribution in duodenum 6 h post intraperitoneal injection of [^3^H]oleate in intralipid (n = 4, 20-week-old female mice). Data represent mean values ± SD; *p* < 0.05 (*), *p* ≤ 0.01 (**), *p* ≤ 0.001 (***); (**e**–**g**) Student’s unpaired *t*-test.

**Figure 4 cells-10-02619-f004:**
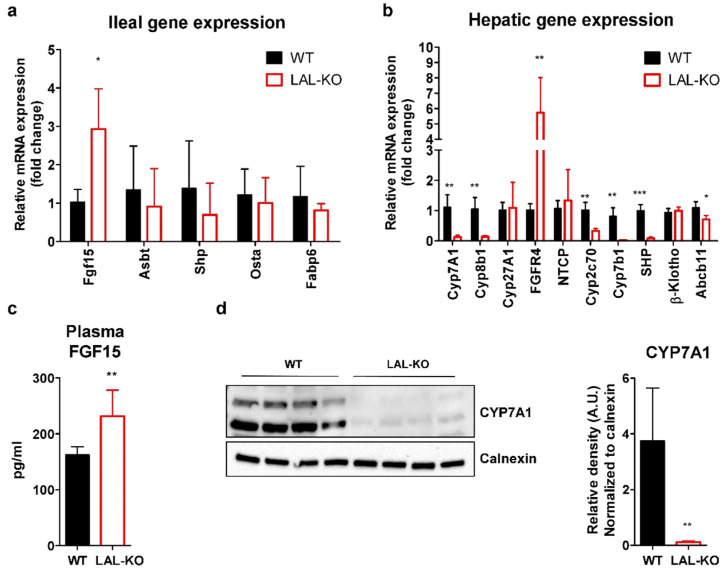
Altered FGF15 signaling in LAL-KO mice: Relative transcript levels of (**a**) ileal and (**b**) hepatic genes relative to *cyclophilin A* as reference gene (n = 4). (**c**) Plasma FGF15 concentrations in WTD-fed mice (12–14 weeks old) fasted for 12 h (n = 7–8). Hepatic protein expression of (**d**) CYP7A1, (**e**) nuclear TEFB, and (**f**) phosphorylated and total ERK including quantification to their loading controls. (**g**) Plasma C4 concentrations (n = 6). Data represent mean values + SD; *p* < 0.05 (*), *p* ≤ 0.01 (**), *p* ≤ 0.001 (***); Student’s unpaired *t*-test.

**Figure 5 cells-10-02619-f005:**
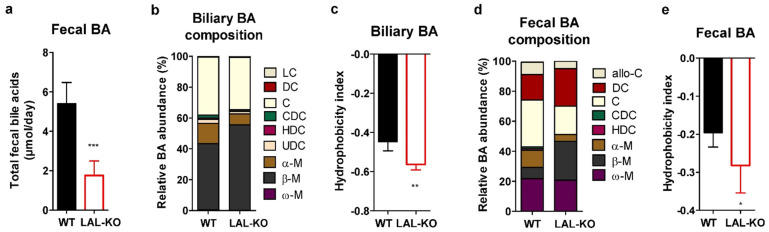
Altered bile acid composition in WTD-fed LAL-KO mice: (**a**) Total bile acid levels in feces, bile acid composition in the (**b**) gallbladder, and (**d**) feces. Heuman’s hydrophobicity index of (**c**) biliary and (**e**) fecal bile acids of WTD-fed male mice (12–14 weeks of age). Data represent mean values + SD (n = 4–6); *p* < 0.05 (*), *p* ≤ 0.01 (**), *p* ≤ 0.001 (***); Student’s unpaired *t*-test.

**Figure 6 cells-10-02619-f006:**
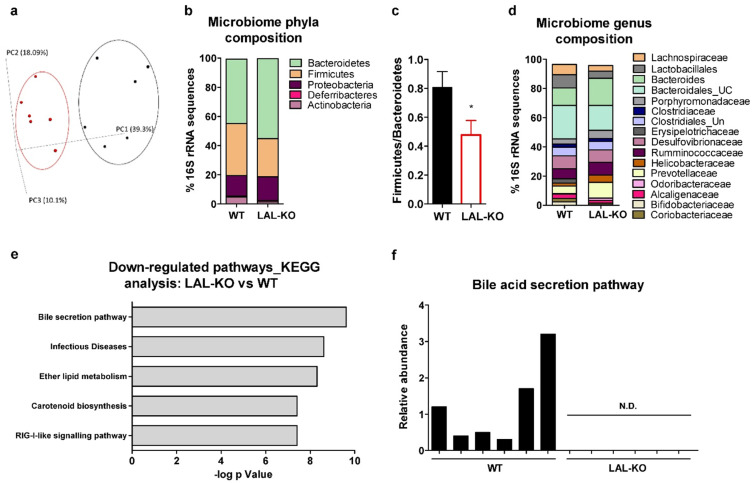
Pronounced shift in cecal microbial communities in LAL-KO mice: Cecal contents of WTD-fed male mice were extracted, and gut bacterial communities were analyzed by 16S rRNA sequencing (n = 6). (**a**) Principal component analysis (PCoA) of groups (WT, black; LAL-KO, red) denote the separation of the colonic microbial community. (**b**) Phylum-level changes, showing average percentages of each phylum as a proportion of the whole community based on genotype; only phyla with relative abundance >0.5% in at least one sample are displayed. (**c**) Ratio of Firmicutes to Bacteroidetes phyla. (**d**) Genus-level changes, showing the average percentage of each genus as a proportion of the whole community based on genotype. For simplicity, only genera with relative abundance >0.1% are displayed. UC, unclassified; UN, unknown. (**e**) Top 5 downregulated biochemical pathways in LAL-KO vs. WT mice based on KEGG analysis. (**f**) Metagenomic modeling using Phylogenetic Investigation of Communities by Reconstruction of Unobserved States (PICRUST) reveals pathways enabling bile secretion. Data represent mean values + SD; *p* < 0.05 (*), (**c**) Student’s unpaired t-test.

## Data Availability

The data presented in this study are available on reasonable request from the corresponding author. Reagents and detailed methods of all procedures are provided in “Material and Methods” of this manuscript or cited accordingly.

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
