# Peer review of "Impaired Bile Acid Metabolism and Gut Dysbiosis in Mice Lacking Lysosomal Acid Lipase"

_cells, 2021, doi:10.3390/cells10102619_

Round 1

Reviewer 1 Report

Major points

- General consideration

The authors seem to make a causative link between what appens in the liver (decrease of expression of enzyme involved in the synthesis/metabolism of bile acids) and what happens in the intestine (alteration of the bile acid pool composition and dysbiosis) which is not convincing. It is probably not possible to determine what is the egg and the hen, this should be clarified, especially in the abstract and the discussion.

The authors state several times that the paper shows the importance of « LAL-derived lipolytic products » in maintaining gut-liver crosstalk, but this is speculative.

- figure 1C, energy expenditure : are the difference statistically significant ?

- figure 1d : as the body weight of the mice is different in WT vs LAL-KO, organ weights should be expressed as percentage of the body weight.

- figure 2 : part is the experiments have been performed in males fed a WTD, whereas parts are performed in females fed a chow diet. These results cannot be gathered in this way. Is there not a sexual dimorphism for cholesterol absorption ? It is likely that the WTD will change the cholesterol absorption and intestinal gene expression.  

- As bile acids constitute a large part of the paper, the analysis should be more extensive. What is the BA profile in plasma ? As the microbiote is altered, the secondary vs primary BA ratio could be changed. It would be informative to have the free and conjugated forms of each bile acid species.

- The hepatic cyp2C70 gene expression is decreased in LAL-KO mice vs WT. As this enzyme converts the CDCA to MCA, how this decrease can be reconciliated with the increase in MCA ?

- An important limitation of the study is the lack of translation of the mouse results to humans, as the major change in BAs is in the MCA, which is absent in humans. The difference in the BA pool between mouse and human should be considered and discussed.

- Limits of the work should be stated in the discussion, mentioning the LAL-KO mouse models does not reproduce all the features of the human disease.

Minor points

- figure 1d : why the weight of the PGAT and not the other AT depots ?

- Figure 2j : title is « enterocytes », it is probably « intestine tissue» or « intestinal epithelium » but not « enterocytes ». Which part of the intestine ?

- figure 4 : the title is not appropriate.

- Heuman hydrophobicity index establishment should be mentioned in the material ad method section.

Reviewer 2 Report

This work from Sachdev V et al. is an interesting study. The experimental design is appropriated and the experimental data perfectly support the conclusion. This is a well conducted study.

This paper is suitable for publication in Cells. If available, I would suggest to include liver BA concentration and BA composition to further support the data showing reduced BA synthesis.

Minor corrections:

Fig. 1c. The authors should clarify the unit of the indirect calorimertry experiment. Is it Kcal/h per mouse ? or KCal/h per g or kg of body weight ? The number of animals per group in this experiment should also be more cleary specified.

Discussion:

Line 503: In my opinion the sentence "The increased fecal neutral sterol loss and fecal lipid excretion together with a 34% reduction in Npc1l1 expression in our study suggested  impaired intestinal cholesterol absorption in LAL-KO mice" is an over-statement. Increased fecal loss of lipids is rather linked to the reduced BA pool. The impact of reduced liver BA synthesis on fecal lipid loss has been extensively demonstrated in the litterature namely in CYP7A1 KO mice or with BA resins. I do not believe that the moderate reduction in NPC1L1 expression could have a major impact on the observed phenotype in LAL KO mice (fecal lipid loss).

Reviewer 3 Report

Dear authors,

After the review process, I have several comments: you should insert numerical data in the abstract; you should clearly present the aim of the paper in the abstract; you should include references in all Materials and Methods sections; in page 13 (second paragraph), you should expand their comments related to LAL deficiency, microbiota, and inflammation processes. Thus, you could make a correlation based on the link between obesity, microbiota dysbiosis, and neurodegenerative pathogenesis. This aspect is important because you should include a future valorization of their research and a study limitation.

Best regards.

Round 2

Reviewer 1 Report

The manuscript has been considerably improved. There are still some changes to be introduced in Supplementary Figure S1.
The terminology of bile acid names is well established and should be used.
CL: what does it mean?
In the figures, the distinction should be made:
- between free bile acids (free BAs, namely CA, CDCA, amMA....), glyco-conjugated bile acids (glycoconjugated-BAs, namely GCA, GCDCA, GUDCA, ...) and tauro-conjugated bile acides (Tauro-conjugated-BAs, namely TCA, TCDCA, TaMCA, ...)
- between primary bile acids (CA, CDCA, aMCA) and secondary bile acids (DCA, LCA, UDCA, bMCA, wMCA)

y-axis name : "BA and conjugates" is not appropriate. "BA" is very large. It should be replaced by "free- and conjugated-BAs".

The standard deviation should be added.

References 34 and 57 are the same one.

Author Response

Reply to Reviewer 1

The manuscript has been considerably improved.

Answer: We sincerely thank the Reviewer for appreciating the modifications and suggesting the necessary changes.

There are still some changes to be introduced in Supplementary Figure S1.

Answer: All requested changes have been inserted in Figure S1.

The terminology of bile acid names is well established and should be used.

Answer: Done as requested.

CL: what does it mean?

Answer: We thank the Reviewer for noticing the typo; CL has been changed to CA.

In the figures, the distinction should be made:

- between free bile acids (free BAs, namely CA, CDCA, amMA....), glyco-conjugated bile acids (glycoconjugated-BAs, namely GCA, GCDCA, GUDCA, ...) and tauro-conjugated bile acides (Tauro-conjugated-BAs, namely TCA, TCDCA, TaMCA, ...)

Answer: Done as requested.

- between primary bile acids (CA, CDCA, aMCA) and secondary bile acids (DCA, LCA, UDCA, bMCA, wMCA)

Answer:  We would like to point out that we have grouped BA into primary and secondary BA based on previous reports [PMID: 23395169 and 31506275]. Primary BA include free and conjugated forms of CA, CDCA, ά-MCA, and β-MCA, whereas secondary BA include DCA, LCA, ω-MCA, UDCA, and their conjugates. We have now included this in the Materials and Methods section, line 204-207.

y-axis name : "BA and conjugates" is not appropriate. "BA" is very large. It should be replaced by "free- and conjugated-BAs".

Answer:  Done as requested

The standard deviation should be added.

Answer:  Done as requested

References 34 and 57 are the same one.

Answer: We thank the Reviewer for this thorough review, we have now corrected this mistake. 

Reviewer 3 Report

Dear Authors,

No other comments.

Author Response

Thank you very much.